# A First Individual-Based Model to Simulate Humpback Whale (*Megaptera novaeangliae*) Migrations at the Scale of the Global Ocean

Jean-Marc Guarini [1,*] and Jennifer Coston-Guarini [2]

1 LEMAR, Brest University, CNRS, IRD, Ifremer, 29280 Plouzane, France
2 The Entangled Bank Laboratory, 11 Rue Anatole France, 66650 Banyuls sur Mer, France
* Correspondence: jean-marc.guarini@univ-brest.fr

**Abstract:** Whale migrations are poorly understood. Two competing hypotheses dominate the literature: 1. moving between feeding and breeding grounds increases population fitness, 2. migration is driven by dynamic environmental gradients, without consideration of fitness. Other hypotheses invoke communication and learned behaviors. In this article, their migration was investigated with a minimal individual-based model at the scale of the Global Ocean. Our aim is to test if global migration patterns can emerge from only the local, individual perception of environmental change. The humpback whale (*Megaptera novaeangliae*) meta-population is used as a case study. This species reproduces in 14 zones spread across tropical latitudes. From these breeding areas, humpback whales are observed to move to higher latitudes seasonally, where they feed, storing energy in their blubber, before returning to lower latitudes. For the model, we developed a simplified ethogram that conditions the individual activity. Then trajectories of 420 whales (30 per DPS) were simulated in two oceanic configurations. The first is a homogeneous ocean basin without landmasses and a constant depth of −1000 m. The second configuration used the actual Earth topography and coastlines. Results show that a global migration pattern can emerge from the movements of a set of individuals which perceive their environment only locally and without a pre-determined destination. This emerging property is the conjunction of individual behaviors and the bathymetric configuration of the Earth's oceanic basins. Topographic constraints also maintain a limited connectivity between the 14 DPSs. An important consequence of invoking a local perception of environmental change is that the predicted routes are loxodromic and not orthodromic. In an ocean without landmasses, ecophysiological processes tended to over-estimate individual weights. With the actual ocean configuration, the excess weight gain was mitigated and also produced increased heterogeneity among the individuals. Developing a model of individual whale dynamics has also highlighted where the understanding of whales' individual behaviors and population dynamic processes is incomplete. Our new simulation framework is a step toward being able to anticipate migration events and trajectories to minimize negative interactions and could facilitate improved data collection on these movements.

**Keywords:** baleen whales; migration; behavior; bioenergetics; individual-based modelling

## 1. Introduction

The distribution of the humpback whale (*Megaptera novaeangliae*) meta-population and their extended migrations have been the object of many studies, but a unified mechanistic framework to explain these movements has not been identified [1]. Statements like "Most baleen whales undertake migrations between low-latitude breeding grounds and high-latitude feeding grounds" [2] confound process and pattern and generate a confusing representation in the literature: is migration a pattern (i.e., an emergent property of individual movements) or a process (i.e., a dedicated set of activities performed to reach a pre-defined destination from a given starting point)? Some (e.g., [3]) have offered explanations of migration in evolutionary terms, stating the question as: "why do baleen whales



migrate?". Corkeron and Connor [3] suggested that migration increases ecological fitness by predator avoidance, hence increasing both survival and reproductive success. Their argument was contradicted by Clapham [4]; even if they agreed that migration is linked to fitness, they contradicted most of the reasoning in [3] and concluded with "However, I acknowledge that convincing explanations for migration await far better data than we currently possess". Migration is also presented as a fitness advantage, but finding strong evidence of this requires long term records of individuals [5]. Migration may then generate large inequalities among individuals that can have consequences on the population dynamics. This led Alves et al. [6] to suggest that the balance between energetic losses and gains during migration could explain these inequalities.

Humpack whales are known to reproduce in fifteen breeding zones distributed in the tropical oceans which are identifed as 14 distinct population segments (DPSs) of the species [7]. Demographic models of whales have been developed (e.g., [8]) and studied extensively (e.g., [9]). Population models have been used to investigate predator–prey and competition processes (e.g., [10]) or ecosystem dynamics (e.g., [11]). Species distribution modeling has also been carried out to account for abiotic environment differences (i.e., habitat) and individual ranges of physiological tolerance. Whale habitat distribution modeling was also the object of species conservation studies (e.g., [12–14]). In parallel, individual-based models have been developed to study particular processes, such as prey–predator interactions (e.g., [15]), whale–ship interactions (e.g., [16]), acoustic impact studies (e.g., [17]) and biological (e.g., [18]), bioenergetic (e.g., [19]) and eco-toxicological dynamics (e.g., [20]).

Migrations though are mostly observed and not modeled. Indeed, many different methods have been developed to observe whale migrations. Some use individual identification [21,22], others used tagged individuals followed by satellites (e.g., [23–25]), genetic markers [26] or stable isotope forensics [27]. Nonetheless, in all cases, studies consist of following a very small number of individuals compared to the size of the population. Thus, observation periods are mostly partial and made in local transit situations.

Inferring population properties from these observations is unreasonable, even if baleen whale populations are made up of a limited number of individuals. This is because individual whales can explore large parts of their habitats. For humpback whales , their habitat is the Global Ocean and individual behaviors should predominate over demographic processes for control of population dynamics [28]. When applying these criteria, individual-based approaches become a good choice for simulating the population dynamics of the species. However, for studying migrations, an additional difficulty arises from the lack of comprehensive knowledge about the behavior and environmental perceptions of whales ([1,28]).

The literature is quite incomplete concerning the sensory capabilities of large marine mammals. While many different mechanisms of biological navigation have been suggested, no conclusive demonstrations are known ([29,30]). One way to consider the problem is that to travel a route directed to a specified destination over long distances and in a constantly changing environment, whales would require both a very good positioning tool and an accurate mapping system. Biological mechanisms for navigation have been described in other animal groups (e.g., [31]). However in an individual-based model, where no specific biological mechanism is invoked, the minimum assumption is that perception is limited (local) relative to the migration distances covered. This permits generation of simple patterns of displacement trajectories under a minimum number of constraints.

Given the partial nature of observations available on whale migrations, modeling migration using a Global Ocean framework offers several advantages. First, mechanistic models can test hypotheses about emergent patterns independently of data and suggest new lines of inquiry. Second, the process of model construction can identify more precisely where conceptual gaps exist. Finally, quantification of interactions between environmental changes, individual behaviors, physiological condition and population dynamics constitutes a step toward being able to forecast ecological outcomes. In this context, studying

whale movement trajectories with simulations contributes to building a comprehensive understanding of links between whales' individual behaviors and population dynamic processes. By using a Global Ocean framework, simulations could inform other efforts to minimize negative interactions with human activities. For instance, the ability to place whale migrations in a Global Ocean means predicted movements could be connected with other information, such as maritime traffic or military activities.

The immediate objective of our study is to test a new model designed to explore if humpback whale migration trajectories can emerge from individual local behaviors. An Individual-Based Model (IBM) was developed that represents the variations of speed and swimming direction as a function of the local depth. The existence of a biological mechanism of depth detection is inferred from acoustic studies ([32,33]); however, this capability remains controversial in the literature. Nonetheless, for a model that presumes bathymetry and behavior interact to prevent stranding, some type of depth detection hypothesis is necessary. A simple ethogram of four behaviors is used to simulate individual behavior dynamics. This includes a feeding behavior to account for fulfilling energy needs along the route of a long distance migration. The dynamic energy budget, including energy acquisition depending on diving events and the food availability is included . Migration route patterns were then studied with two different topographic configurations: one with a homogeneous ocean and without land masses and a second using the actual bathymetry and coastlines. Finally, the ecophysiological profiles of the groups of individuals within each DPS were also characterized in both configurations.

## 2. Materials and Methods

The model in the article was designed to simulate the dynamics of humpback whale individuals at the scale of the Global Ocean. It uses the 1 arc-minute global relief topography (ETOPO-1) provided by NOAA [34], including the presence of ice sheets covering bedrock in Greenland and Antarctica. To account for the Arctic and Antarctic summer Ice Packs, two physical barriers for whales were set at 75° N and 70° S, respectively. The ETOPO-1 topography is available in two registration formats, called "grid" and "cell". The grid registered format was used here because it is more accurate; the cell registered file is built by resampling the grid registered file and topographic relief may be flattened locally. Using the grid format permits further development, such as for connecting the model with a hydrodynamic framework. The size of the resulting table is 21,601 values in longitude values by 10,801 values in latitude values, with duplicated values at the North and South Poles and on the stitching longitude line (180° E and 180° W).

### 2.1. Humpback Whales' Shape, Size and Growth

Humpback whales have been primarily described by their sizes: the total length, $L$ (in m), surface, $S$ (in m$^2$), volume, $V$ (in m$^3$), and weight, $W$ (in kg). To simplify the simulation framework, it was assumed that their growth is isometric. Hence the body volume and weight are functions of $L^3$ and the surface is a function of $L^2$, $V^{2/3}$ and $W^{2/3}$. The isometric coefficient, $a_{WL}$ linking the weight to the length, described by [35], was re-estimated with an exponent 3 instead of 2.95. The fineness was assumed to be constant and set to 5.0 [36,37]. This was used with length to calculate the surface and volume of each individual whale. Three simplified geometric shapes were tested: the first is made of two cones stitched at their identical base [35], the second one is made of two paraboloids stitched as their identical base and the third is a prolate ellipsoid. Results were compared with the a relationship linking surface and weight [38] and with the estimated body density [39,40].

The generic isometric individual growth model is:

$$\dot{W} = gW^{2/3} - rW \tag{1}$$

where $g$ in $M^{1/3} \cdot Time^{-1}$ is the weight growth rate and $r$, in $Time^{-1}$ is the weight loss rate.

The growth in length is formulated as:

$$\dot{L} = \gamma(L_\infty - L) \tag{2}$$

where the growth rate, in $Time^{-1}$, is $\gamma = r/3$ and the asymptotic size, in $m$, is $L_\infty = ga_{WL}^{-1/3}/r$.

It was assumed that the length, $L$, which depends on the growth of the bones (i.e., skull and skeleton), was ensured in any conditions. The corresponding metabolic cost was included in the basal metabolism.

The simplified ecophysiological individual model formulated here combines the growth and the dynamic energy budget of the organism, distinguishing the bones from the storage compartment (i.e., the blubber) and the remainder of the body (see top diagram in Figure 1). The individual dynamics of these three compartments were formulated as a System (systems of equations are indicated by the capital "S" in the remainder of the text) :

$$\begin{cases} \dot{W}_S &= g_S W_S^{2/3} - r_S W_S \\ \dot{W}_B &= \sigma W_O - dW_B \\ \dot{W}_O &= a W_O^{2/3} - u_M W_O - s W_O + \delta W_B \end{cases} \tag{3}$$

where $W_S$, in *Mass*, is the weight of the bones, $W_B$, in *Mass*, is the weight of the blubber and $W_O$, in *Mass*, is the weight of the rest of the body. The total weight, $W$, is then equal to $W = W_O + W_B + W_S$. The bone accretion rate, $g_S$, is in $Mass^{1/3} \cdot Time^{-1}$. The bone growth regulation rate, $r_S$, is in $Time^{-1}$. The food source assimilation rate, $a$, is in $Mass^{1/3} \cdot Time^{-1}$. The weight loss rate, $u_M$, is in $Time^{-1}$ and corresponds to the physiological activity of the organism. Energy storage rates are, $s$, in $Time^{-1}$ and $\sigma$, in $Time^{-1}$, and energy release rates are $\delta$, in $Time^{-1}$ and $d$, in $Time^{-1}$.

### 2.2. Simplified Energy Budget

This section details processes used in the ecophysiological model, System (3), as represented in the middle diagram of Figure 1. The first state is the bone weight. The isometric growth, in length, is described by Equation (2). The rates $r_S$ and $g_S$ were defined based on the results of this equation:

$$r_S = -\frac{3}{T_\omega} ln\left(1 - \frac{L_\omega - L_{min}}{L_\infty - L_{min}}\right) \tag{4}$$

where $L_{min}$, in $m$, is the size at birth, $L_\infty$, in $m$, is the maximum size, $L_\omega$, in $m$, is the size at weaning and $T_\omega$, (in *days*), is the time between birth and weaning.

Then, $g_S$ can be calculated as:

$$g_S = \frac{3\zeta_S^{1/3} r_S L_\infty}{a_{WL}^{-1/3}} \tag{5}$$

where $\zeta_S$ is the optimal proportion of bone weight in the total weight of the whale individual. Indeed, the dynamics of bone weight, $W_S(t)$, do not depend on the dynamics of the two other state variables, $W_B(t)$ and $W_O(t)$. According to the assumption made for the growth of bones, it only refers to an optimal body size. However, to ensure that the approach remains consistent, the metabolic cost of the bone growth, $u_S = r_S W_O/W_S$, was compared to the basal metabolic rate $u_b$, which should be higher in any condition.

The two other states, $W_B(t)$ and $W_O(t)$, are interdependent. The link is primarily controlled by the balance between weight gains and losses. The weight loss rate, $u_M$, is decomposed as the sum of the basal metabolic cost, $u_b$, the metabolic cost of swimming, $u_m$, and the cost of feeding $u_f$:

1.  The basal metabolic rate, (in $Energy \cdot Time^{-1}$), was quantified by [41] as a function of $W^{(3/4)}$. Therefore, $u_b$ was formulated as:

$$u_b = \frac{C_b}{\xi_O} W_O^{-1/4} \tag{6}$$

    where $\xi_O$, in $Energy \cdot Mass^{-1}$, is an energy density coefficient for the $W_O$ compartment. The basal metabolic coefficient, $C_b$ is in $Energy \cdot Mass^{-3/4} \cdot Time^{-1}$.

2.  The metabolic cost of swimming, $u_m$, was formulated according to Hind and Gurney [42]:

$$\begin{cases} D_m &= 0.50\rho\left(a_{SW}W^{-1/3}\right)C_D V_m.^2 \\ u_m &= 86400\frac{\lambda}{\epsilon_a\epsilon_p}\frac{D_m V_m}{\xi_O} \end{cases} \tag{7}$$

    where $\rho$, in $Mass \cdot Length^{-3}$, is the volumetric mass of seawater, $C_D$, *dimensionless*, is a drag coefficient characteristic of the species, $V_m$ in $Distance \cdot Time^{-1}$ is the swimming speed and $a_{SW}$ is the coefficient linking the wet surface to the total weight of the humpback whale individuals. According to [38], the wetted surface is, on average, about 85 % of the total surface of the individuals.

3.  The feeding cost, $u_f$, is the sum of the diving cost and the lunge cost ([19,43]), respectively, $u_d + u_l$. $u_d$ was estimated to be $k_d = 3.75$ times the basal metabolic rate and $u_l$ was described as :

$$u_l = \frac{C_l}{\xi_O} W_O^{1/3} \tag{8}$$

    where $C_l$, in $Energy \cdot Mass^{-4/3} \cdot Time^{-1}$, is a lunge cost coefficient.

Following the reasoning above, the assimilation rate, $a$, depends on resources ingested during foraging. This required introducing (in System (3)) an additional variable, $W_R$, *in Mass*, representing the mass of food ingested in the digestive system:

$$\dot{W}_R = I - \alpha W_O^{2/3} \tag{9}$$

where $I$, in $Mass \cdot Time^{-1}$, is the ingestion rate during feeding and $\alpha$, in $Mass^{1/3} \cdot Time^{-1}$, is an ingested-to-assimilated mass transfer rate. $I$ was expressed as:

$$I = 3.5a_{EL}L^3[R] \tag{10}$$

where $a_{EL}$ (dimensionless) is a coefficient of the function linking the length of the humpback whales to their engulfment volume [43], assuming that, on average, 3.5 lunges are performed for each 10 min dive. $[R]$ is the concentration of resources within the volume engulfed. $\alpha$ was expressed by:

$$\alpha = \frac{\iota_R}{L^2} W_R \frac{\xi_R}{\xi_O} \tag{11}$$

where $\iota_R$, in $Time^{-1}$, is a digestion rate and $\xi_R$, in $Energy \cdot Mass^{-1}$ is the energy density of the resources.

The assimilation rate, $a$, is thus equal to $a = \alpha\kappa_a$, where $\kappa_a$, dimensionless, is the assimilation efficiency. The mass loss in excrement per time is equal to $(1 - \kappa_a)\alpha W_O^{-2/3}$; this term implicitly subtracted matter from System (3).

Humpback whales feed on krill and small pelagic forage fishes [44]. To quantify the energy requirements of the whale, it is necessary to quantify how many resources are ingested during feeding and how much energy is spent, not only for foraging but also for any other task that whales accomplish. During foraging, the energy spent for diving is predictable (hence, losses were deterministic), but the energy acquired may be much more variable, depending on the quantity and quality of food that whales encountered and could engulf. In our preliminary approach to the question, $[R]$ and $\xi_R$ were considered to

be both constant and non limiting. However, this description could be modified if a global description of whale prey spatio-temporal patterns became available in the future.

Processes that control the dynamics of storing and releasing matter and energy from the blubber are controlled by the balance between gains and loss. This balance, $\Delta$, is expressed as:

$$\Delta = \alpha \kappa_a W_O^{2/3} - u_M W_O \tag{12}$$

Then , if $D > 0$, the rates $s$ and $d$ are formulated as:

$$\begin{cases} s &= \dfrac{2\Delta}{3}\dfrac{1}{W_O} \\ d &= 0 \end{cases} \tag{13}$$

Otherwise, if $D \leq 0$, the rates $s$ and $d$ are formulated as:

$$\begin{cases} s &= 0 \\ d &= \dfrac{-\Delta}{W_B \xi_B / \xi_O} \end{cases} \tag{14}$$

where $\xi_B$, in $Energy \cdot Mass^{-1}$, is the energy density coefficient for the blubber compartment ($W_B$). In addition, $\delta = d(\xi_R/\xi_O)$ and $\sigma = s(\xi_R/\xi_O)$, where $\xi_R$, in $Energy \cdot Mass^{-1}$, is the energy density coefficient for the resources ($R$).

### 2.3. Seasonal Activity Cycle and General Ethogram of Humpback Whales

Behavior refers to actions and reactions of whale individuals to internal cues or external stimuli. Because whales are mobile organisms, many behaviors are associated with movements. There is, however, a persistent confusion in the definition of behaviors regarding the term "migration (e.g., [1]). Like Friedlaender in 2013 [45], we also found no comprehensive ethogram of humpback whale behavior in the literature. We developed an original ethogram (Figure 1) from reviewing the literature on whale behaviors (Table 1), to simulate the behavioral dynamics of individuals. Overall, six behavior categories were defined for the ethogram: Resting, Exploring, Feeding, Traveling, Breeding and Interacting (Table 1).

**Table 1.** Definition of the humpback whale behaviors in the model ethogram. Horiz. Mov. and Vert. Mov. stand for horizontal and vertical movements respectively.

| Behavior | Common Definition | Sources | Horiz. Mov. | Vert. Mov. |
|---|---|---|---|---|
| Resting | characterized by low activity. Individuals stay at the water surface or just below. They perform little or no visible movements of appendages (pectoral fins, fluke), except for exhalations | [1,45–49] | null | null |
| Traveling | results in relatively large-scale horizontal displacements characterized by a persistent directional heading (high autocorrelation and small turning angles) and a sustained speed; surface active behaviors are low, or infrequent | [28,45,47–49] | fixed | null |
| Feeding | involves diving and various rapid sequences of swimming movements described as lunges, jerking and off-axis rolling motions associated with deceleration when the whale's mouth is open | [28,45,50] | null | fixed |
| Exploring | refers to general movements including random horizontal displacements and dive series to gather information about the biotic and abiotic environments | [45] | variable | variable |
| Breeding | is a set of behaviors resulting in producing offspring: breeding involves mating with multiple individuals (males and females) | | null | null |
| Interacting | any presumed communication interaction either at distance (i.e., sound production) or with direct encounters between individuals; includes all social and agonistic behaviors | [45,49,51] | null | null |

We then assumed that behaviors representing movements would be linked with swimming and could define the term "migration" as only referring to a long distance displacement by individual organisms. In other words, a migration is an emergent property of the swimming behavior. We explicitly do not include terms like "cyclic", "periodic" or "seasonal" for the model because we do not want to impose an assumption or constraint of pre-determined destination zones and areas of residence. This permits us to explore how the population is organized in space and time based only on individual organisms' local environment perceptions. Swimming was divided into two distinct behaviors [45], the first, called Exploring is characterized by random horizontal directions, a variable speed and frequent diving at variable depths. The second, Traveling, is characterized by a fairly constant direction and sustained averaged speed, mainly at the sea surface. Even if the traveling behavior predominates during a long distance journey, it is not exclusive and may occur when individuals tend to stay in a small area.

Some ancillary remarks are needed to complete the definitions presented in Table 1 regarding the model design:

- Resting is a fundamental behavior allowing not only to recuperate from efforts produced when swimming and foraging [48] but also to preserve energy before migrating [46], this being crucial for the survival of calves during the upcoming migration. Whales have a particular way of sleeping which was described by Lyamin et al. ([52–54]) and quantified by [55]. This typical uni-hemispheric sleep covers all sleeping stages [56] and is estimated to last, on average, 40% of the duration of the day [54].
- Exploring includes both horizontal exploration of an area and a vertical exploration of the water column, down to deep levels [48], to find suitable depth for feeding.
- Feeding is restricted to a behavior characterized by diving followed by prey foraging [19]. Prospecting dives without prey foraging are included in the exploring behavior. This implies that prey were assumed to be present when Feeding was performed in the simulations.
- Traveling is characterized by swimming at a fairly constant route and speed. This does not mean, in our model, that the destination was planned in advance, even in the case of migration. Traveling can occur outside of the migration period [45], with identical characteristics, while route and speed during Exploring were considered to be random.
- The next two categories are not included in the model at this time because there is not enough information on their energetics available:

  Breeding covers all the many behaviors related to sexual reproduction of the whale, from adult intercourse to caring for calves. It therefore interacts with all other behaviors and seasonal cycles of activity [57]. Breeding behaviors are also associated with sound production and are a part of the whale acoustic seascapes [58]. Since there is no new individual produced in our model, Breeding was not explicitly accounted for in the present study.

  Interacting overlaps with: Breeding, Feeding and Traveling. Energetic costs can be direct or indirect, negative, neutral or positive and are always difficult to quantify. No group dynamics were formulated in the model. Interacting was not explicitly represented in the study, even if individuals get close to each other from time to time in the simulations (Figure 1).

The ethogram matrix crosses only four modalities (of the six in Table 1): $b_i = Resting$, $Traveling$, $Exploring$, $Feeding$ with themselves, $b_j$, to establish Table 2) used in the simulations.

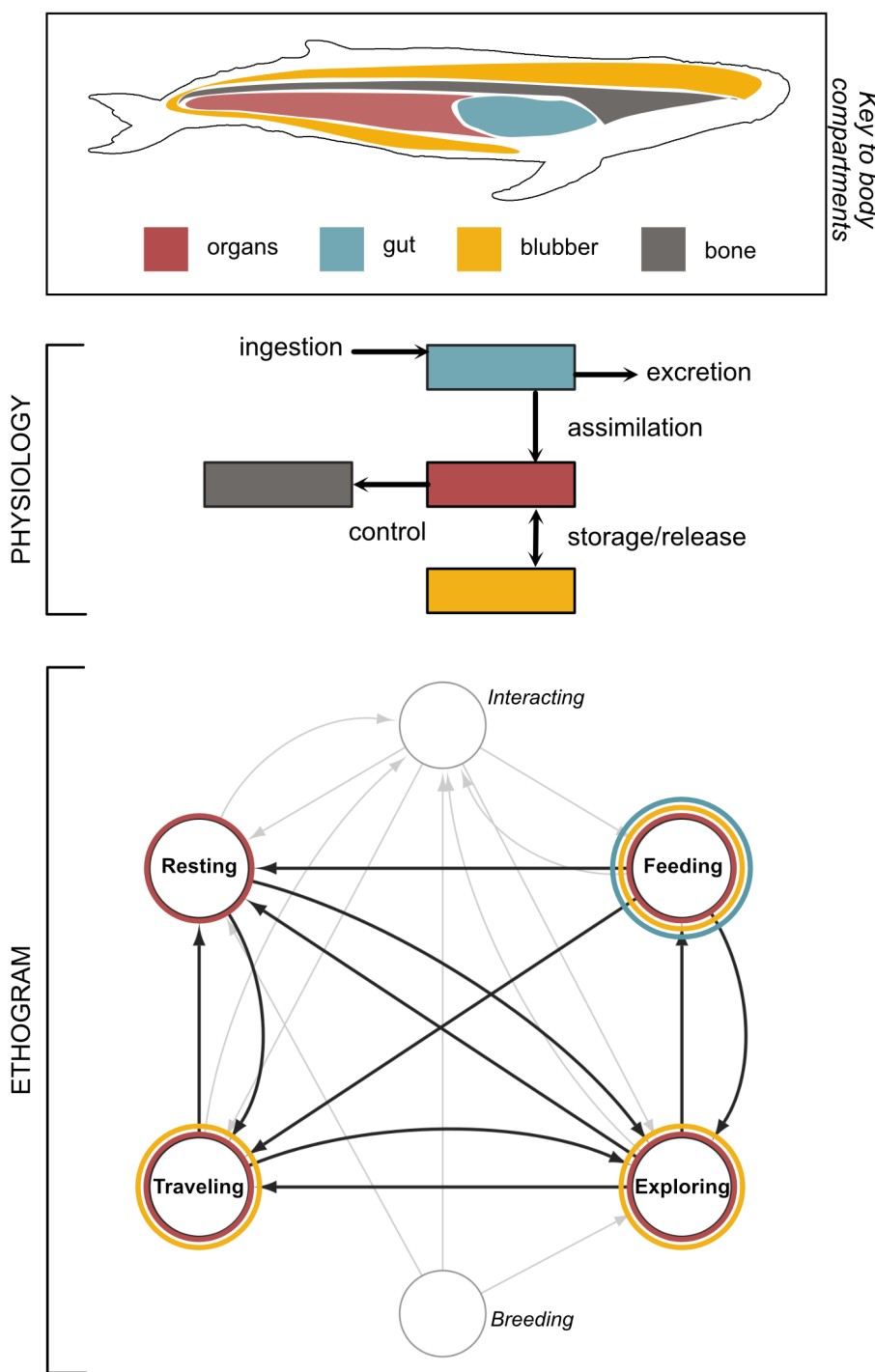

**Figure 1.** Concepts manipulated in the individual-based model framework. Each whale individual is described by two components: 1. The physiology and bioenergetics were described by exchanges between four body compartments, gut, organs, blubber and bone. 2. A generic, comprehensive ethogram was defined for the humpback whale based on the literature. Solid lines and arrows indicate the links between four behaviors treated by the model. The graph also indicates (in gray), two other behaviors that were not considered due to the lack of suitable energetic information. The ethogram graph was transformed into a transition matrix that defines the probability of maintaining one behavior or changing from one to another during simulations.

**Table 2.** Ethogram Matrix, with $E_b = 1$ when modalities are linked, $E_b = 0$ when modalities are not linked and $E_b = R$ for the retention ($bi = bj$).

| From\To | Resting | Traveling | Exploring | Feeding |
|---------|---------|-----------|-----------|---------|
| Resting | R | 1 | 1 | 0 |
| Traveling | 1 | R | 1 | 0 |
| Exploring | 1 | 1 | R | 1 |
| Feeding | 1 | 0 | 0 | R |

The resulting graph is plotted in Figure 1, as a subset of the full ethogram (of six behavioral categories). It was implicitly stated, at the beginning of this section, that whale individuals have different phases of activity, during which all behaviors can occur but with different relative importance. The first difference among activities is between long distance displacements or residing in the same area. The second level of difference is that, in the low latitude residence areas, during winter, individuals may enter into a phase of sexual reproduction. In order allow for a seasonal pattern of activity, the relative orientation of the Earth toward the Sun was computed as a function of the latitude and declination, regardless of the hour angle (i.e., the cosine of the hour angle is equal to 1):

$$\begin{cases} dec = 23.45 sin\left(\dfrac{2\pi}{365.25}\left(j + 284.0 + j_{lag}\right)\right) \\ \Psi = sin(dec)sin(\phi) + cos(dec)cos(\phi) \end{cases} \tag{15}$$

where $j$ is the day of the year, $j_{lag}$, the phase of the lagged declination ($dec$), $\phi$ is the latitude of the whale. The index that represents how whales perceive this general environmental change is the first derivative of $\Psi$. This index is consistent with the hemispheric phase inversions of the seasons. The transition between the phases of residence and migration are triggered by an arbitrary threshold parameter defined such that no changes in activity occur at the equator. From the ethogram matrix (Table 2) was calculated a transition matrix using the table of cumulative daily durations for each of the four behaviors, $T_b$ and for the four different activity regimes (Table 3):

**Table 3.** Table of the cumulative daily behavior durations, $T_b$, hours.

| Activity\Duration | Resting | Traveling | Exploring | Feeding |
|-------------------|---------|-----------|-----------|---------|
| 0 (30° S–30° N) | 16.0 | 0.5 | 5.0 | 2.5 |
| +1 (Equator → Pole) | 4.5 | 16.0 | 1.0 | 2.5 |
| 0 (High Latitudes) | 14.0 | 0.5 | 4.5 | 5.0 |
| −1 (Pole → Equator) | 4.5 | 16.0 | 1.0 | 2.5 |

From the durations in Table 3, the $4 \times 4$ transition matrix (crossing the four modalities $b_i = \{Resting, Traveling, Exploring, Feeding\}$ with themselves) was calculated for each individual and for each of the four activity regimes:

- 0 means the activity occurs while residing in a latitude between 30° S and 30° N;
- +1 is when it is moving from equator to poles;
- A second 0 value is possible while residing in high latitudes (greater than 30° N or 30° S);
- −1 means the whale is moving from one of the poles toward the equator.

Finally, the probabilities at the intersection of the crossed modalities varied with time and were calculated as:

$$\begin{cases} p_{bi->bj}|_{i=j} = \dfrac{T_b - t_c}{T_b} \\ p_{bi->bj}|_{i\neq j} = \dfrac{t_c}{T_b}\dfrac{E_{bi->bj}}{\sum_{j=1}^{4}(E_{bi->bj})} \end{cases} \tag{16}$$

where $t_c$ is the time (i.e., $t = t + dt$), where $dt$ is the simulation time step, being limited by $T_b$.

### 2.4. Movement of the Humpback Whale Individuals

The behaviors of the whales were represented explicitly. Whales move horizontally when Traveling or Exploring only. Their significant vertical movements (i.e., diving during Exploring or Feeding) were not represented explicitly. Horizontal movements are described by a direction (defined clockwise with $0°$ heading to the north) and a swimming speed. The rules applied in the model are:

1   Regarding horizontal movements, an average direction is provided as a function of the perception of the local environmental gradient. When the gradient is perceived as positive (increase of environmental variables, as, for example, sea surface temperature), individual whales move toward the pole. In contrast, when the gradient is perceived as negative, whales are moving toward the equator. In the open ocean, the average swimming speed was equal to 1.6 m·s$^{-1}$ ([59,60]).

2   Whales' perception of the environment is local, limited to nine cells, hence 31 km$^2$. In shallow waters, (depth lower than 200 m), whales were assumed to perceive changes in the bathymetry and adjust their route in consequence. The speed decreases and the route changes to avoid stranding. The decision rule to change route is to take the closest suitable direction in one of the seven sectors around the original heading.

3   When exploring in open oceanic areas, whales moved according to a correlated random walk. The speed was on average equal to 0.5 m·s$^{-1}$ [61]. Speed is highly variable between 1.0 m·s$^{-1}$ and 0.0 m·s$^{-1}$ when Exploring involved diving. In shallow waters, the random mode is canceled and the route is adjusted to avoid stranding.

4   During foraging, the horizontal speed decreased down to zero, whales dived and engulfed resources. Diving was not represented explicitly. Whales were assumed to optimize both the diving effort and time spent to ingest food at the foraging depth. According to [62], a dive at a maximum of 120 m deep (average 80 m) lasts typically 10 min during which up to 3.5 lunges can be performed, depending on the densities of the preys.

5   The resting behavior is represented in a minimal way; during the resting time, swimming speed is assumed to be equal to zero and the physiological processes are resumed to the basal metabolism.

An important ecological implication of the traveling behavior rules is that, when the heading is constant, routes followed are loxodromic. This implies that the environment at the surface of the water was assumed perceived by whales as flat. In other words, the rotundity of earth is not perceived by whale individuals.

### 2.5. Parameters and Conditions of the Simulations

To quantify the dynamic energy budget, an individual whale was described as a prolate spheroid, parameterized with two basic dimensions, the length (in m) and the diameter (in m), corresponding, respectively, to the longest dimension and the largest width of the shape. The parameters used in the model are regrouped in Table 4.

The model simulated the movements of 420 whales, 30 individuals per DPS. The initial positions of the whales (Figure 2) were randomly withdrawn within the known distribution area of their respective DPS [7]. Then successive positions of the whales were calculated independently from each cell. It was necessary to calculate the correspondence between them at each time step (i.e., determining constantly the cells in which the whales were located) to define the local environment that they were assumed to perceive. The cell dimension is 1 square arc-minute. Therefore, cell size varies with latitude since the longitudinal dimension is a function of the cosine of the latitude. Thus, the whale individual's perception of the environment decreased in size with longitude when whales moved toward higher latitudes and vice versa. Each individual's state was described by a position, a heading, a speed and a set of four weights (in tons) corresponding to the gut content, the bone mass,

the blubber mass (reserve of fat) and the rest of the soft body. The food source was not limited, equal to 3.0 kg·m$^{-3}$, any time and anywhere whales performed feeding. The time step of the simulation was constant and equal to 6 min. This duration was selected to avoid that whales skip adjacent grid cells when moving. Simulations were performed for a total duration of 24 months. Their seasonal activities were driven by the local gradient perception and their behaviors were determined by withdrawing a probability in their behavior transition matrix. Their movements, bioenergetic rates and weight dynamics were calculated at each time step.

The model code was computed in gfortran (gnu compiler collection (gcc), version 4.4.3). Simulation results were treated in Scilab 6.1.1 and Datagraph 5.0.1. The ethogram network is graphed in Cytoscape 3.9.1.

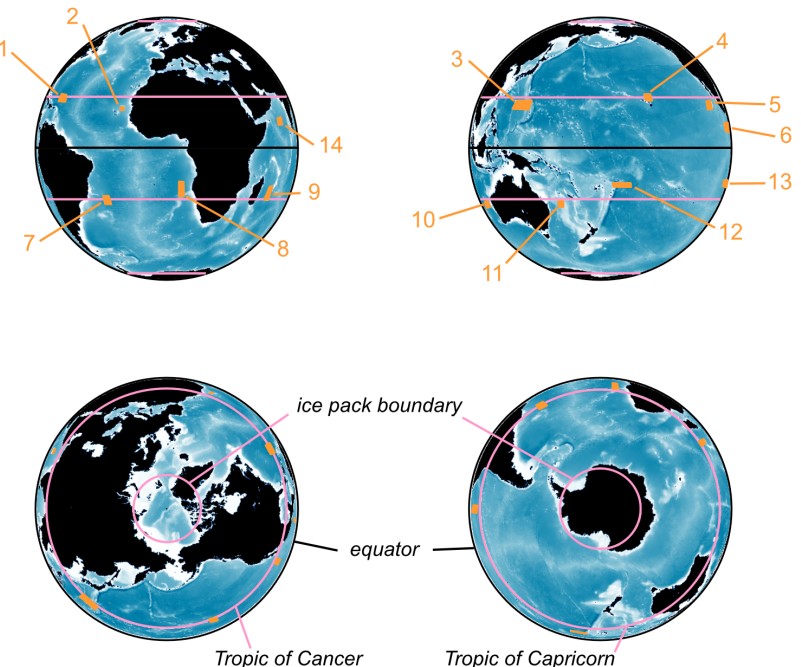

**Figure 2.** Configuration of the simulated Earth system and positioning of the Distinct Population Segments, numbered from 1 to 14: 1—West Indies, 2—Cabo Verde, 3—Western North Pacific, 4—Hawaii, 5—Mexico, 6—Central America, 7—Brazil, 8—Gabon and Southwest Africa, 9—Southeast Africa and Madagascar, 10—West Australia, 11—East Australia, 12—Oceania, 13—Southeastern Pacific, 14—Arabian Sea. Five parallels structure the space: the equator, the tropical latitudes and the summer limits of the Ice Packs in the Arctic (75° N) and Antarctic (72° S).

**Table 4.** Table of the parameters used for numerical simulations.

| Symbol | Signification | Unit | Value |
|---|---|---|---|
| $L_{min}$ | minimum length | m | 4.00 |
| $L_\infty$ | maximum length | m | 17.00 |
| $L_\omega$ | length at weaning | m | 8.50 |
| $t_\omega$ | age at weaning | days | 320 |
| $\zeta_S$ | bone to total weight ratio | dimensionless | 0.11 |
| $\zeta_B$ | blubber to total weight ratio | dimensionless | 0.25 |
| $\zeta_O$ | organs to total weight ratio | dimensionless | 0.64 |
| $f$ | fineness | dimensionless | 5.00 |
| $a_{WL}$ | coefficient | kg·m$^{-3}$ | 14.46829 |
| $a_{EL}$ | coefficient | dimensionless | 0.00540 |
| $a_{SW}$ | coefficient | m$^2$·kg$^{-2/3}$ | 0.06208 |
| $\gamma_L$ | length growth rate | day$^{-1}$ | 0.0017 |

**Table 4.** *Cont.*

| Symbol | Signification | Unit | Value |
|:------:|:-------------:|:----:|:-----:|
| $C_b$ | basal metabolism coefficient | $J{\cdot}kg^{-3/4}{\cdot}h^{-1}$ | 12,204 |
| $C_D$ | drag coefficient | dimensionless | 0.003 |
| $C_l$ | Lunge cost coefficient | $J{\cdot}kg^{-4/3}{\cdot}h^{-1}$ | 28.62 |
| $\lambda$ | active to passive drag ratio | dimensionless | 0.20 |
| $\epsilon_a$ | aerobic efficiency | dimensionless | 0.15 |
| $\epsilon_p$ | propeller efficiency | dimensionless | 0.85 |
| $k_d$ | BMR to DMR multiplying factor | dimensionless | 3.75 |
| $\iota_R$ | digestion rate | $h^{-1}$ | 1.0 |
| $\kappa_a$ | assimilation efficiency | dimensionless | 0.75 |
| $\xi_O$ | energy density coefficient for the $W_O$ | $kJ{\cdot}kg^{-1}$ | 18,500 |
| $\xi_B$ | energy density coefficient for the $W_B$ | $kJ{\cdot}kg^{-1}$ | 38,000 |
| $\xi_R$ | energy density coefficient for the $R$ | $kJ{\cdot}kg^{-1}$ | 4500 |
| $j_{lag}$ | phase of the lagged declination | days | $-40$ |

## 3. Results

### 3.1. The Whales' Bathymetric Environment

The bathymetric component of the ETOPO1 global relief data (Figure 2) constrained the movements and dynamics of the whales. The proportion of negative height is 0.661 for the "Ice Sheet" file used in this study, and 0.708 for the corresponding "Bedrocks" file. The average negative height is $-3435.77$ m for the "Ice Sheet" file and $-3233.96$ m for the corresponding "Bedrocks" file. The minimum height is $-10,898$ m in both cases. The dimensions of the ocean found with the "Ice Sheet" file is $361 \times 10^6$ km$^2$ and $368 \times 10^6$ km$^2$ for the "Bedrocks" file. The calculated volume is $1.3328 \times 10^9$ km$^3$ for the "Bedrocks" file and $1.3358 \times 10^9$ km$^3$ for the "Bedrocks" file. The $75°$ N parallel and $72°$ S parallels were set as an Ice Pack barrier preventing whales swimming further toward the respective poles. The calculated summer Arctic Ice Pack area with a southern limit at $75°$ N was equal to 7.144 km$^2$ and the Antarctic one with a northern limit at $72°$ S was equal to 2.117 km$^2$ using the "Ice Sheet" data set. These calculated areas were equal to 7.341 km$^2$ and 6.791 km$^2$ for summer Arctic and Antarctic Ice Packs, respectively, using the "Bedrocks" data set.

### 3.2. Quantitative Description of the Whale Shape

The lengths of the humpback whale individuals were assumed to range between 4 m (length at birth) and 17 m (asymptotic length). Re-fitted allometric relationships for weight vs. length and surface vs. weight led to average differences equal to ca. 0.2 tons and ca. 1.0 m$^2$, respectively. It was considered to be small enough to assume that the isometric growth hypothesis was valid. The simplified shape model retained was a double-paraboloid stitched at the base, assuming a ratio of their height $h_1/h_2$ = ca. 3 [63]. This shape provided estimates of surface and density closest to estimates from [38] for the surface without appendages (fins and flukes) and from [39,40] for the body density. It is worth noting that the surface estimated with a double-paraboloid equaled the surface estimated from [38] when $h_1/h_2$ = ca. 14.

### 3.3. Seasonal Activity and Migration Patterns

The local time variations of the gradient index of environmental change perception are shown in Figure 3, for seven different latitudes ($75°$ N to $75°$ S with a $25°$ step). The plots show how it oscillates between high positive values and low negative values (indicating that the global environment variable decreases). Perception of the temporal changes (i.e., a local gradient) was assumed to vary also with latitude. First, the amplitude of the oscillation increases with latitude, with a maximum value equal to ca. 0.404 at $90°$ N and $90°$ S. The two threshold values which separated periods of activity were fixed at 0.100 and $-0.100$. Thus, between these two thresholds, the local environmental gradient is considered to be non significant and individuals would not exhibit any sustained oriented movement.

Above 0.100 or below −0.100, the gradient is considered to be locally significant. Hence, individuals move consistently toward higher latitudes if index values are positive and toward lower latitudes if index values are negative. At the equator, the threshold is never reached; the maximum and minimum index values were 0.081 and −0.081, respectively. At the equator, the frequency of the index doubled, exhibiting two oscillations per year.

The calculation of the index also shows an inversion of phase between the northern and Southern Hemispheres, accounting for the inversion of the seasons. The perception of the gradient is local and depends on the position of the whale. The resulting activity pattern is characterized by alternating periods of residence in either lower or higher latitudes and periods of migration in between.

The movements of 30 individuals in each of 14 Distinct Population Segments (DPSs) were calculated over a 2-year period. All 420 Individuals had the same initial weight equal to 14.5 tons. Two conditions were simulated. The first corresponded to a virtual, homogeneous ocean covering the entire Earth, with a constant depth equal to −1000 m. There was no emerged land, hence no coastline. Only the two boundaries made by the two Ice Packs regions at the poles were maintained. The second condition is a more realistic configuration of the ocean, with a variable bathymetry and presence of land masses and coastlines, as estimated by the ETOPO-1 data.

Figures 4 and 5 show the variations of the activity and daily behaviors in the constant depth configuration (Figure 4) and actual Earth topography (Figure 5) for all 420 whales simulated in both hemispheres. All DPSs have similar environmental conditions and only differed in their initial localization. There are two fundamental levels of temporality. The upper plots show the variability of the activity index defined by the perception of the environmental gradient; the activity index was equal to +1 when changes were significantly perceived as positive, hence triggering travel toward high latitudes; it was equal to −1 when changes were significantly perceived as negative, hence triggering travel toward low latitudes; and it was 0 when changes were not significantly perceived, allowing the subject to reside in the local area). Even though seasons are not actually specified by the model, these figures show activity patterns emerge that coincide with seasonality in each hemisphere, with a small variability among individuals.

The lower plots in Figures 4 and 5 summarize the occurrence patterns of each behavioral category for the whales. Each vertical mark indicates the occurrence of the behavior at a time point. Behaviors change dynamically at the time resolution of the hour (i.e., the simulation time). Changes were modulated by the transition probability, accounting for the average time spent in each of the behavioral categories. The two categories "Resting" and "Feeding" are behaviors that occurred every day for all individuals in both bathymetric configurations. In both sets of results, "Traveling" is associated with a more seasonal activity pattern, decreasing to zero for all individuals during the periods in which they reside either in feeding or breeding grounds. "Exploring" is a much scarcer behavior, and shows slight increases during residence periods.

However, using actual bathymetry and coastlines, activity patterns fluctuated more than in the constant depth simulations (Figure 4). This is due to the larger dispersion in the local positions (primarily in the latitudinal direction) of the whales (e.g., Figure 6). In particular, individuals situated in the Arabian Sea (DPS14) had their migration duration halved in simulations with the actual bathymetry configuration because their route is blocked by the Asian continent. For some groups, particularly in Central America (DPS6), in southeastern Africa (DPS9), in eastern Australia (DPS11) and in Oceania (DPS12), the temporal variability was high with some individuals exhibiting short migrations while others traveled long distances between their respective breeding ground and feeding ground residence locations.

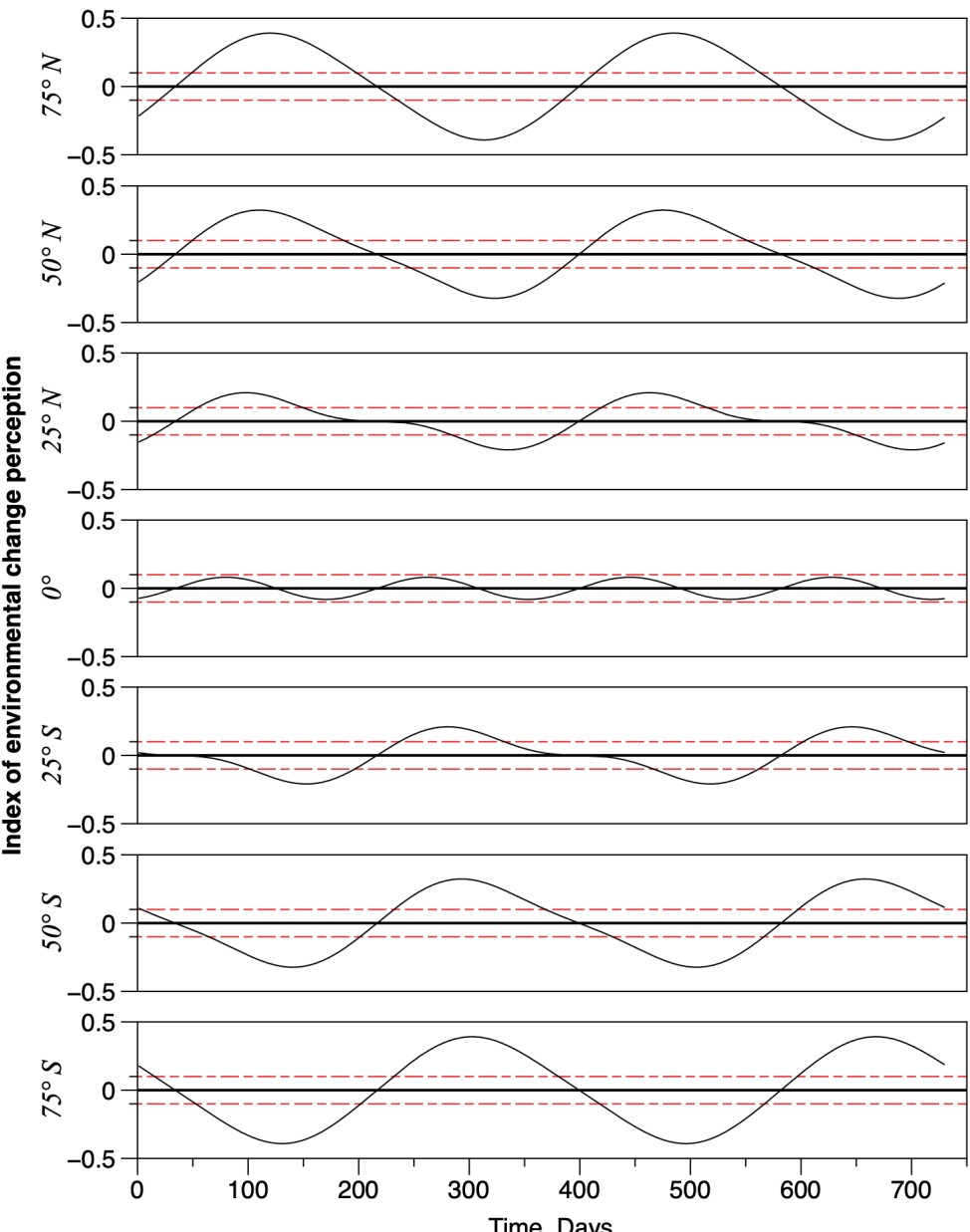

**Figure 3.** Examples of the spatio-temporal fluctuations of environmental changes as perceived by individuals in the model. The black curves represent the variations of the index of perceived changes. The red dashed lines are the 0.1 and −0.1 thresholds, above and below which environmental change perceptions are significant (positively and negatively, respectively). The black solid line indicates the zero baseline. Seven latitudes are represented, showing the doubling of the oscillation when converging to the equator. In addition, at the equator, there is no significant perception of environmental changes.

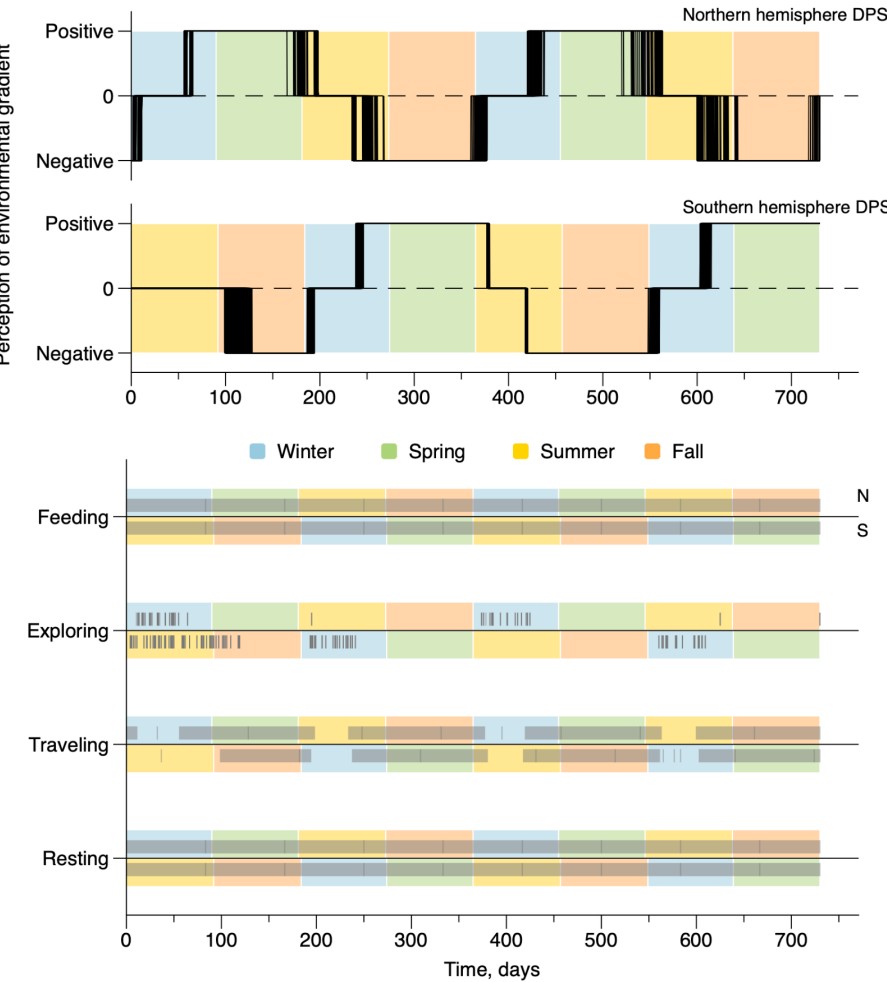

**Figure 4.** Time variations of the activity and behavior of whales in the constant depth with no coast configuration. The upper figure represents the realized perception of the environmental gradient defining the activity, superimposed on a seasonal cycle at mid-latitudes. When the perception was positive, whales migrated from low latitudes toward the poles, when the perception was negative, whales migrated from high latitudes toward the equator. The lower figure shows the occurrences (black lines) of the four behaviors superimposed on a seasonal cycle at mid-latitudes.

Overall, the dynamic patterns in behaviors were very similar with the one recorded in the constant depth ocean. The "Exploring" behavior is more frequent during residency in breeding or feeding grounds. This was due to the higher variability in the migration period duration. From the series of activities and behaviors, a table of realized cumulative average, daily behavior periods ($\tau_b$) was back-calculated for the groups of DPS of both the Northern and Southern Hemispheres and in both topographic configurations. The values were not significantly different, so we have reported only one value in each cell of Table 5.

**Table 5.** Table of the cumulative daily behavior period duration, $\tau_b$.

| Activity/Duration | Resting | Traveling | Exploring | Feeding |
|---|---|---|---|---|
| 0 (30° S–30° N) | 21.0 | 0.0 | 0.0 | 3.0 |
| +1 (Equator → Pole) | 8.9 | 11.5 | 0.0 | 3.6 |
| 0 (High Latitudes) | 19.0 | 0.0 | 0.0 | 5.0 |
| −1 (Pole → Equator) | 8.9 | 11.5 | 0.0 | 3.6 |

The back-calculated probabilities, as a result of the simulated dynamics, are consistent but do not match with the initial duration table (see Table 3). Relative to the earlier table,

the "Resting" and "Feeding" behavioral categories occur more (longer durations), while "Traveling" and "Exploring" occur less (shorter durations). This difference is because of the dynamics of sequential behaviors in a stochastic context.

The perception of the environmental gradient is local, but sustained oriented movements (i.e., traveling behavior) were fast enough to maintain a long distance journey. This feature is more critical for the movements from the poles to the equator because the perception of the environmental gradient tends to decrease both in intensity and duration when approaching lower latitudes (Equation (4); Figure 3).

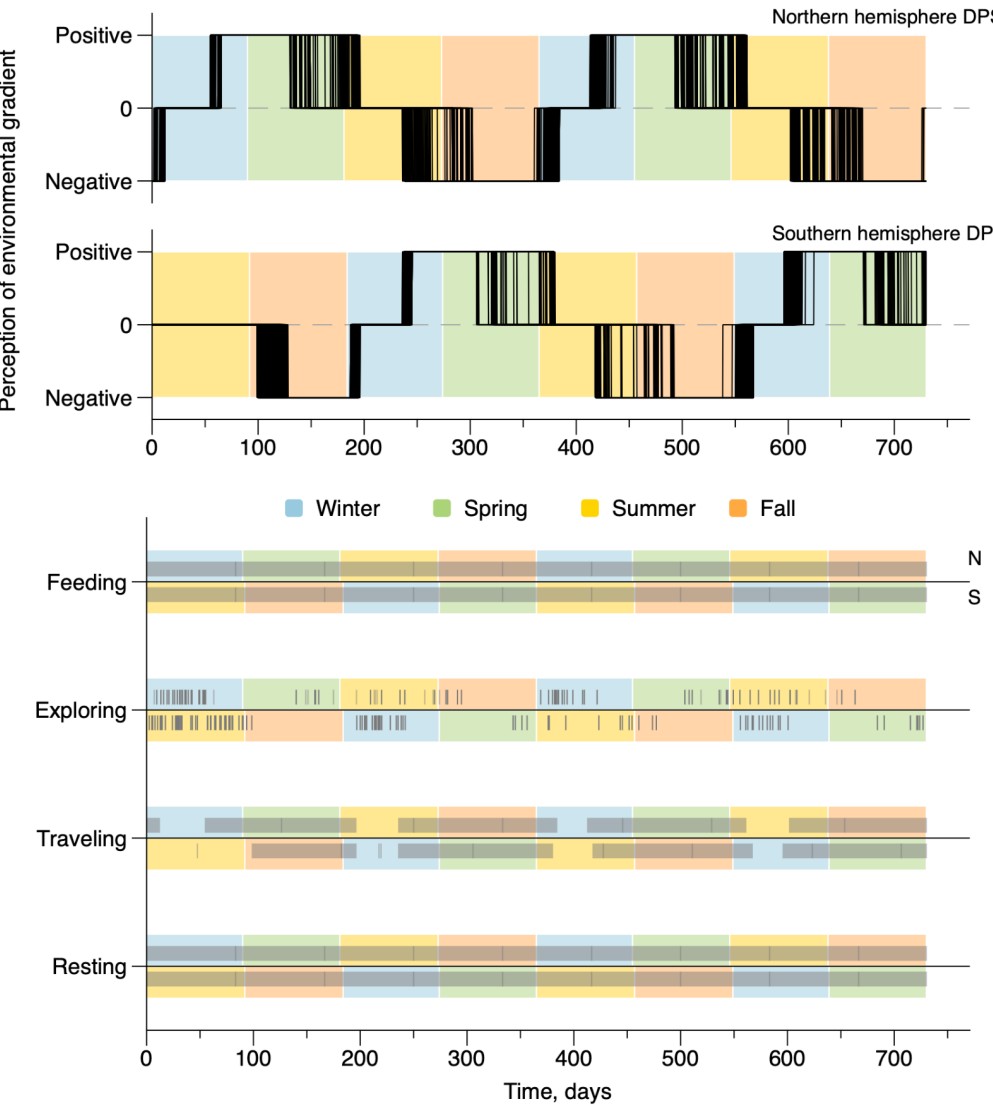

**Figure 5.** Time variations of the activity and behavior of whales in an actual Earth topography configuration. The upper figure represents the realized perception of the environmental gradient defining the activity, superimposed on the seasonal cycle at mid-latitudes. When the perception was positive, whales migrated from low latitudes toward the poles, when the perception was negative, whales migrated from high latitudes toward the equator. The lower figure shows the four behavior occurrences (black lines) superimposed on a seasonal cycle at mid-latitudes.

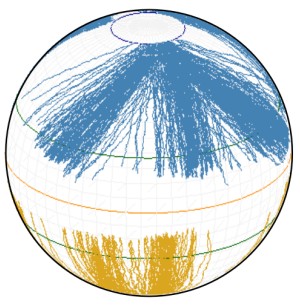 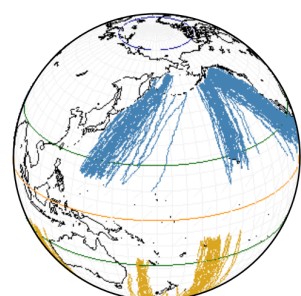

no coastline, constant
depth of −1000 m

with actual coastline and
bathymetry

**Figure 6.** Examples of the routes followed by individual whales from their winter breeding grounds to the higher summer feeding grounds. In blue, the Northern Hemisphere, in orange, the Southern Hemisphere. The left figure shows a simulation performed in a configuration where the depth of the ocean was constant (−1000.0 m) without any emerged land. The right figure shows a simulation performed in the configuration of the actual Earth topography, estimated at 1 arc-minute resolution. The equator (orange line), the tropics (green lines) and the summer boundaries of the Ice Packs in the Arctic and the Antarctic were represented.

Examples of the trajectories followed are plotted in Figure 6, for both globe configurations. All other conditions and parameters were identical for these simulations. The migration pattern is characterized, in both configurations, by long commutations between the tropical zones and the polar regions. However, in the constant depth ocean basin configuration, the back and forth movements tend to largely disperse whale individuals around the globe, blurring the notion of a DPS. Using the actual bathymetric configuration of the Earth, trajectories were more constrained, maintaining distinctive groups in the tropical zones. In the constant depth configuration, the average total distance was ca. 15,000 km·y$^{-1}$, which is about 41 km·d$^{-1}$, at 1.7 km·h$^{-1}$. These characteristics decreased down to 6000 km·y$^{-1}$, which is about 16.5 km·d$^{-1}$, at 0.7 km·h$^{-1}$ in the actual topographical configuration of the Earth. This is due, in our simulations, to the decrease of speed in shallow waters, preventing individuals from stranding.

In both configurations, no individual stranded in the two years of simulations. The latitudinal ranges of the migration remained quite similar: the equator was never crossed and the Ice Packs boundaries were reached in the Southern Hemisphere for most of the DPS. In the Northern Hemisphere, the presence of land obstacles (e.g., the Aleutian Islands in Figure 6) limited access to the Ice Pack boundary, which was only reached by a small fraction of the individuals. The variability in distance and speed between DPS was high in the actual topographical configuration of the Earth. The maximum variability was recorded for the Northern Hemisphere, with minimal values calculated for DPS14, in the Arabian Sea, because movements toward high latitudes are impossible due to the presence of the Asian continent at 25° N.

Figure 7 represents the average variation of depth at the whale's positions for each DPS. The straight line indicates a depth of −1000.0 m, chosen for the constant depth configuration. The fluctuating lines represent the average depth for trajectories performed in the actual bathymetric configuration of the Earth. They varied largely both in time and space. Initial conditions were randomly defined, for most of the cases, in areas where the ocean is deep, at the level of the abyssal plains, around −4500 m. This allowed whales to start with free swimming, without starting avoiding major obstacles. The main exception is the DPS located in eastern Australia, starting in areas at −2000 m. In most of the cases, most of the whale individuals tended to move toward and stay in a much shallower area, around −1000 m, on the continental slopes and in the vicinity of the continental break. A noticeable exception is the group of whales from the West Indies (DPS1), which alternated visits to very shallow and very deep areas; the whales from the Southeastern Pacific (DPS13), which

remained in deep areas; and the whales from the Arabian Sea, which tended to stay in the vicinity of the coasts and on the continental shelf. The trend to converge to very shallow areas was observed for DPS groups located in the Western North Pacific (DPS3) and eastern Australia (DPS11).

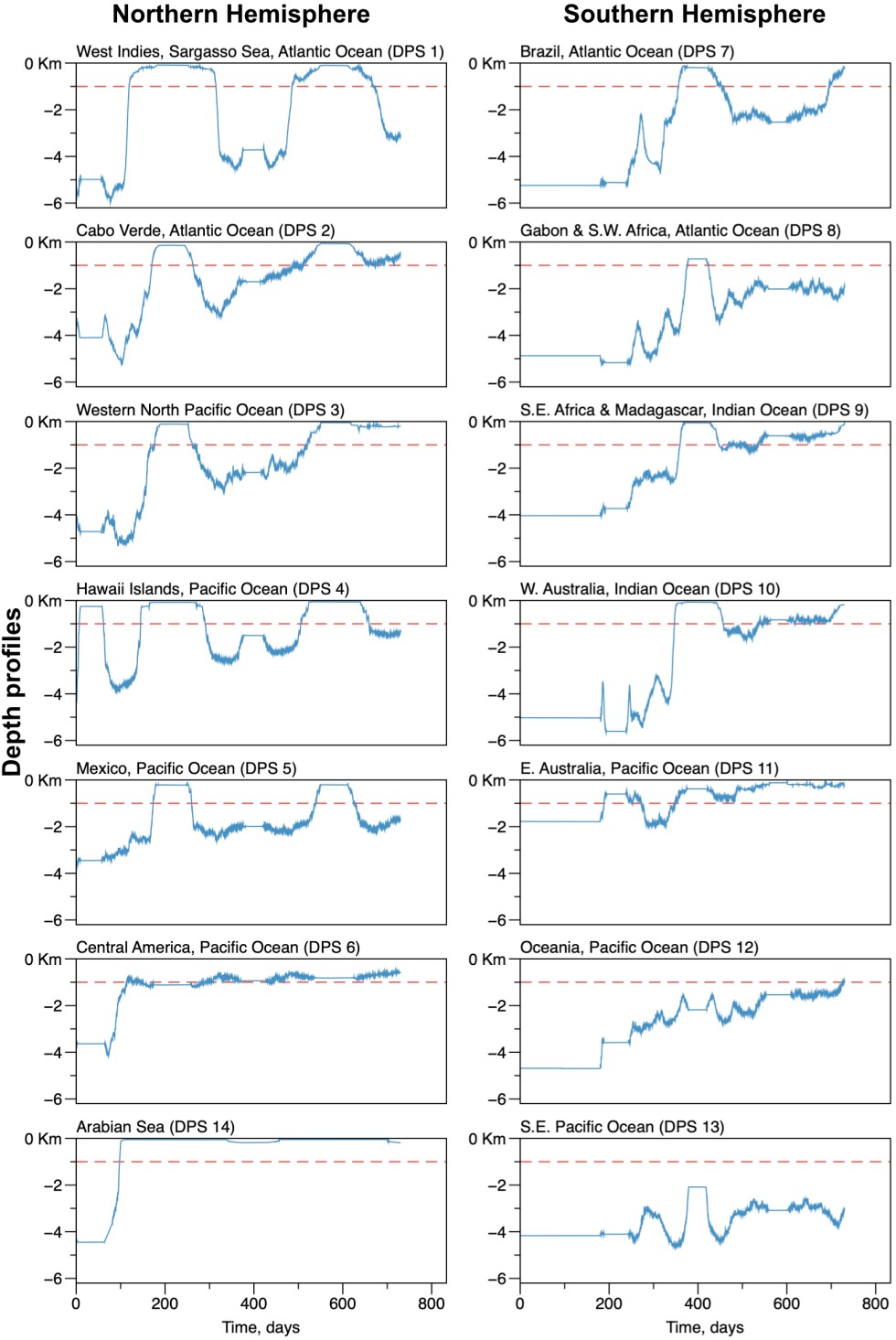

**Figure 7.** Average depth profile for the areas frequented by groups of whales corresponding to each of the Distinct Population Segments. In each figure, the blue line shows the depth profiles in a configuration where the depth of the ocean was constant (−1000.0 m) without any emerged land. The blue line shows the depth profiles in the configuration of the actual Earth topography, estimated at 1 arc-minute resolution.

### 3.4. Related Metabolic Budget and Weight Growth Patterns

The consequences of seasonal activity pattern and behavior dynamics on the physiology and growth of the individuals are presented in Figures 8 and 9. Ecophysiological processes interacted with behaviors, when Feeding and Resting led to increased weight and contributed to storing energy in the blubber, while Traveling and Exploring induced a decrease in weight, triggered by a net increase in energy consumption and release of energy from the blubber (Figure 1). Figure 8 shows the dynamics of growth of an average individual in terms of the average change in total weight of all the organisms (upper curves) for each DPS. The black dotted line represents the nominal, isometric curves (Equation (1)) and the blue (actual ocean bathymetry) and red (homogeneous ocean depth) lines are the simulated total weights, as the sum of the blubber, bones and rest of the body masses (System (3)).

The nominal isometric growth for the total and bone weights are identical for all individuals, regardless of the DPS they belonged to. This is because the physiological rates are assumed to be the same for all individuals. The initial mass for the overall body was equal to 14.7 tons, which corresponds to 1.6 tons for the bones. At the end of the two years, the nominal total mass was equal to 33.9 tons, corresponding to 3.7 tons of bones. The growth rate for the total weight was equal to 0.1647 day$^{-1}$ and, for the bones, 0.0789 day$^{-1}$. The regulation rate for the growth was 0.0040 day$^{-1}$ (for both total and bone weights). The asymptotic weight in the nominal condition is 70.7 tons.

The simulated bone weight dynamics (which is a monotonic growth) were all confounded because the growth was assumed to follow the nominal isometric curve as long as there is enough energy reserve (i.e., that the mass of the blubber did not decrease down to zero), which was always and everywhere the case. The simulated total weight, which is the sum of the constrained bone mass and unconstrained mass of the blubber and rest of the organism, fluctuates differently: alternating slow growth phases during residence in low latitudes (breeding grounds) and during migrations and faster growth phases during residence in higher latitudes (feeding grounds).

In general, the total weight was systematically over-estimated in the configuration of a constant depth ocean. At the beginning of the simulations, as the initial weights were set to the nominal values, the differences were small, but they became significant at the end of the two years. They fluctuate between 59.1 tons, which is 1.5 times the nominal value, in the Southeastern Pacific (DPS 13) and 71.5 tons, which is 1.8 times the nominal value, in Central America (DPS 6). The results were very different with the configuration of the actual Earth topography; they fluctuate between 19.6 tons, which is 0.5 times the nominal value, in the Arabian Sea (DPS 14)—where there is no distinctive feeding ground—and 57.0 tons, which is 1.4 times the nominal value, in the West Indies (DPS 1). For many other DPSs (in particular, 2, 7, 8, 10 and 12) the final total weight was very close to the final values of the nominal growth. For DPSs 3, 4, 6, 9 and 11, the weight growth was slightly underestimated.

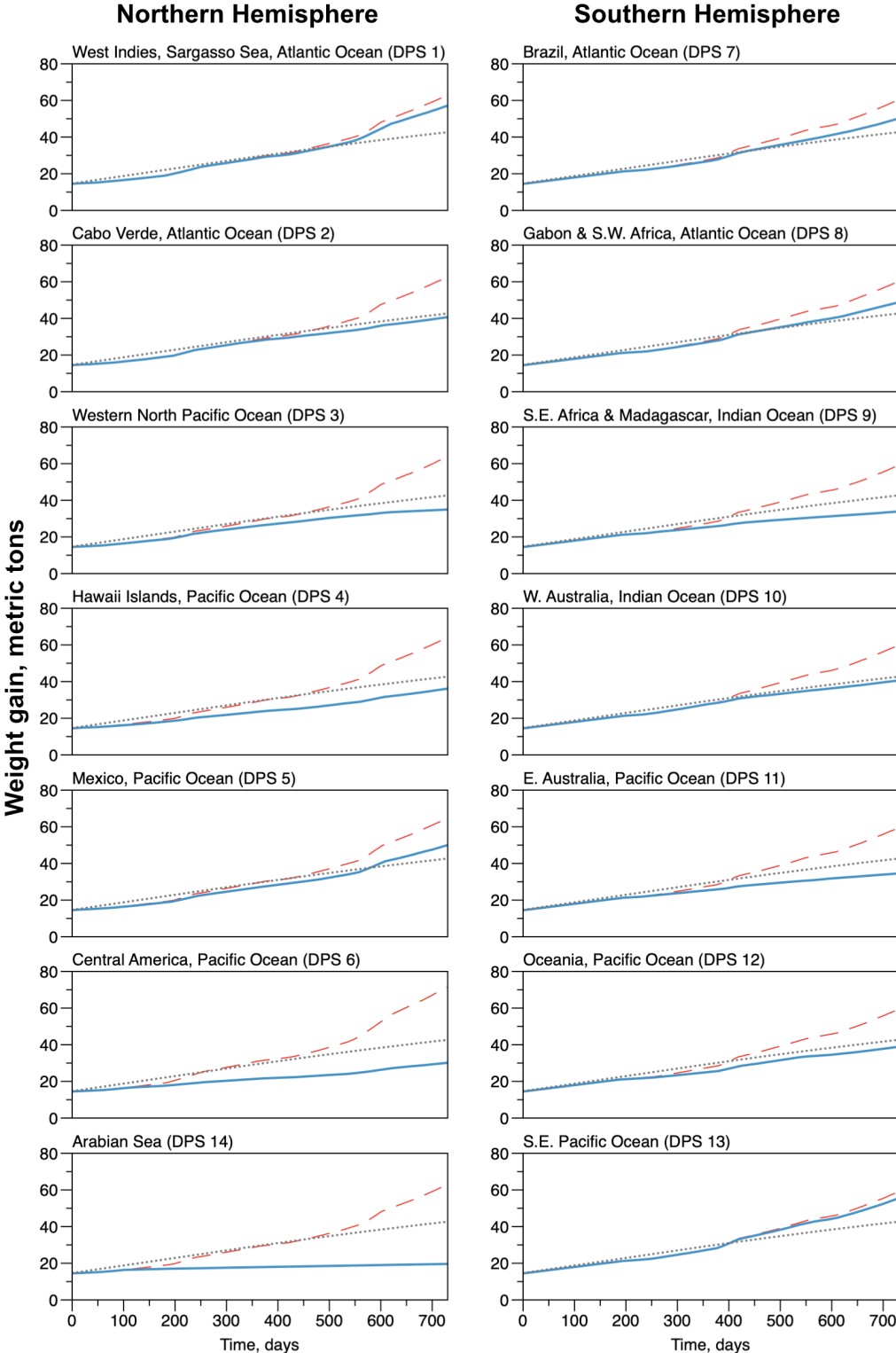

**Figure 8.** Average weight variations for the groups of whales corresponding to each of the Distinct Population Segments. The dashed gray line shows the total weight (upper line) calculated as a nominal isometric growth. Red dashed lines show the total weight evolution in a configuration where the depth of the ocean was constant (−1000.0 m) without any emerged land. The blue lines show the total weight in the simulations with the actual Earth topography, estimated at 1 arc-minute resolution.

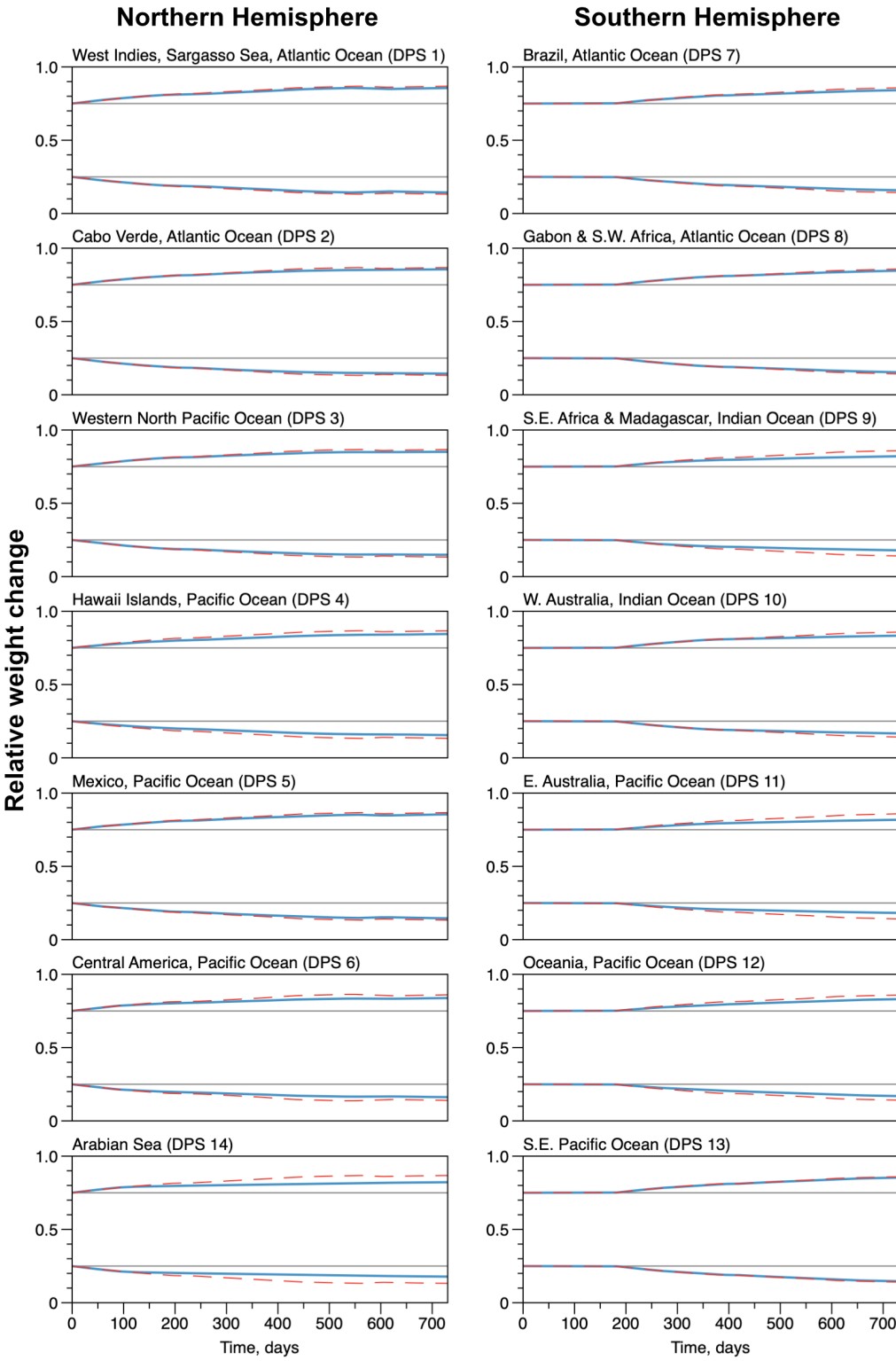

**Figure 9.** Average weight proportion for the groups of whales in each Distinct Population Segment. The lower line groups (below 0.5) are the ratio between the blubber and the total weight. The upper line groups (above 0.5) are the remaining body compartments (organs, bones, gut) divided by the total weight. The red dashed lines result from simulations with the constant ocean depth and no coastlines, the blue solid lines are from simulations with the actual ocean bathymetry. The nominal ratios (0.25 and 0.75) are also plotted as dotted gray lines.

The conditions of the averaged individuals for each DPS are plotted in Figure 9. The condition is the average ratio of the blubber over total weight and of the rest of the body over total weight. The nominal ratios of 0.25 for the blubber and 0.75 for the rest of the body are also indicated (gray lines on plots). In the constant depth ocean configuration, the realized ratio for the blubber decreased down to 0.13, which is about 0.5 times the nominal value, while the realized ratio for the rest of the body increased up to 0.87, which is about 1.16 times the nominal value, for all DPSs. This indicates that the individuals are overweight. The dynamics of the blubber mass is characterized by alternating phases of increases (when individual resided in low and high latitudes) and decreases (when individuals migrated), but the dynamics of the rest of the body monotonously increased, refueled by the blubber energy during migrations. In the configuration of the actual Earth topography, the ratio followed the general trend of the weight; however, it tended to decrease less for the blubber ratio and to increase less for the rest of the body. For the blubber ratio it decreased down to 0.16, which is about 0.65 of the nominal value and the realized ratio for the rest of the body increased up to 0.84, which is about 1.13 times the nominal value, with a small variability between all DPSs. The dynamics for the blubber and for the rest of the body were similar to the configuration of a constant depth ocean, but the decreasing phases of the blubber weight dynamics were more pronounced. The exception was again the group in the Arabian Sea (DPS 14), where the blubber dropped out to a low value in the first days of the simulation and did not recover afterward, while the rest of the body maintained a normal increase initially but almost stopped growing in mass afterwards. In general, for the actual estimated configuration of the Earth, individuals tended to lose energy reserves, even if the individuals maintained their total weight regarding the nominal values of the isometric growth.

## 4. Discussion

### 4.1. Simulating Whales Population—Individual-Based Models Should Be Developed

Very few studies have proposed an integrated individual-based modeling study at the level of population dynamics. Dodson et al. [64] have proposed an IBM model to study the complex behavior of blue whales along the west coast of North America in relation to environmental variables, mainly Sea Surface Temperature (SST) and krill abundance. This model was constructed to explain their observed migration pattern. Our model, even if it shares some characteristics with Dodson et al.'s [64] approach, was designed using different assumptions and choices.

Our model articulates three interlinked components of the humpback whale individuals: their physiology, behavior and seasonality. Their activity was triggered by an external, local stimulus, representing the perception of their surrounding environment change, wherever they are located. We did not use one particular environmental factor, or a combination of them, because this requires quantifying them at the scale of the individual perception and determining the mechanism and the sensitivity of the perception of changes in the factor, and for each individual. The question of what factor(s), exogenous or endogenous, trigger(s) whale migration is difficult to solve considering the current knowledge ([1,65]). Quantification of the physiology of the organism is one of two mandatory components of individual-based modelling [66]. The physiology describes how individuals can act and react, defining how energy is allocated to biological functions [59]. In our study, it was not only a question of quantifying explicitly energy losses and gains, but also of taking into account changes of weight, because this is crucial for medium- and long-term simulations, especially for fast growing young individuals. The second mandatory component of IBM is quantification of the dynamics of the organism behavior. We chose to simulate the behavior dynamics based on a transition matrix in which the dynamics are a stochastic Markov process. This transition matrix is calculated from an ethogram matrix that we defined and the duration table (Tables 2 and 3) modulated by the seasonal activity. However, building the ethogram confronted an increasing complexity in the description of behaviors with a lack of quantitative information about them. The ethogram was therefore restricted to a

set of four behavioral categories. This approach could be improved by linking behaviors with local environmental variables (e.g., the state of the ocean, the state of the atmosphere, the quantity of food, the presence of other individuals) and the physiological conditions of the organisms. For this to be possible, however, much more information is required at the scale of the individual that is both not currently available and difficult to simulate accurately [67].

The model also differs from [64] in that it aims to simulate the dynamic of the overall humpback whale population, at the scale of the Global Ocean. This is, to our knowledge, the first model that has attempted this and we encountered numerous important challenges. First, in 2018, the IUCN estimated that the global whale population was about 135,000 with about two-thirds being mature individuals [68]. The abundance distribution and status of sub-populations is heterogeneous and simulating the dynamics of such a number of organisms remains a significant computing effort. Second, the resolution of the outcome of physical models (coupling ocean, land and atmosphere variables) has become finer but remains far larger than what individuals are expected to perceive (with the exception of acoustical information transmission). Furthermore, the integration of biogeochemical cycles and ecological variables is only beginning and remains preliminary [69]. In end-to-end ecosystem approaches [70], our model can be seen as a point of convergence, defining what is required to introduce populations of the largest marine organisms in which components of behavior and physiology predominate.

### 4.2. The Movements of Whales Are Much More Than Migration

Our model has raised many questions about active movements in the ocean that go beyond the problem of migration patterns. In principle, whales can swim at the surface or under the surface, can breach and can dive to various depths. All this activity was assumed in the model to be based on the local perception of the environment, without knowledge of the position or of the overall configuration of the Global Ocean. Swimming at a fairly constant heading and speed was considered to be a behavior (i.e., "Traveling"). This assumption implies that the whale's perception of the surface of the ocean is flat and the result is that the simulated migration routes are loxodromic. It differs from the long distance migrations of birds, which have been assessed as orthodromic [31,71]. In whales, this assumption is very difficult to verify, first because there are not enough tracks to lead to statistical assessment, second because there is no consensus about navigational cues implied [29,30] and because there are many obstacles that modify and reduce the routes. Kettemer et al. [57] have recorded a long round-trip route between the north coast of Hispaniola Island and the southeast coast of Iceland, performed at 1.6 m·s$^{-1}$, which deviated from the shortest route and which has a shape of a loxodromic route, once the variability is filtered out. However, this constitutes a unique record and does not allow for a conclusive generalization.

In the simulation framework, horizontal movements related to the "Exploring" behavior were considered random. This behavior can occur during all seasonal activity phases of the life cycle, but becomes predominant over "Traveling" in a breeding or feeding ground. Curtice et al. [61] illustrated this exploring behavior with tagged individuals followed in the Antarctic peninsula during an austral summer. The random prospecting of the area was interspersed by short "Traveling" movements, when individuals followed a constant heading and faster sustained speed. An interesting way to consider these straight displacements outside of the migration activity phase would be to define "Exploring"-related movements as a Lévy walk instead of a purely random walk [72]. It would be particularly interesting since the distinction between a Lévy and random walk is linked to the distribution of the resources, that is whether they are dispersed or aggregated. In particular, a Lévy walk is observed when organisms try to find a resource which is distributed sparsely and aggregated [73]. In a more general way, there is a need to link the dynamics of the whale with the dynamics of the resource they consume. This however requires knowing the environmental detection radius for prey of each individual whale. It was considered to be

limited to the eight cells around the cell that contains the individual (hence, a maximum radius of about two arc-minutes, or 3.7 km, decreasing in length with latitude). Volkenandt et al. [44] have suggested that the detection radius of suitable food resources can be up to 8 km, even indicating that only some forage fish species were targeted and others were disregarded. The echolocation capacities of the humpback whales are suggested in the literature [33] as a means to detect prey, without specifying the distances over which this may function effectively.

### 4.3. Seasonal Condition of the Whales

The condition of individual whales was set to vary primarily with the seasonal activity cycle, which modulated the expected daily duration available to execute each behavior. The conditions of all individuals were identical at the start of the simulation, so that the realized performances depend only on the changes in environmental perception. Working with a simulation framework allows manipulation of single factors in the environment to disentangle which causes observed differences.

When simulations had a uniform ocean basin depth and no coastlines, the differences between individuals were small, only driven by the starting position in the DPS locations distributed around the tropics. These differences tended to decrease because of strong dispersion of individuals in this configuration. In all cases using the uniform ocean basin, the model over-estimated the total weight growth, compared to a nominal isometric growth curve. The over-estimation mainly concerned the growth of the organ compartment that consumes energy for the functioning of the organism. This means the compartment has exceeded its mass and energy requirements. In contrast, when the actual Earth topography is used, the simulated weight increase curves are much closer to the nominal weight [35]. The difference is attributed to the effort made to avoid obstacles when individuals were in shallow waters. Individuals whose growth remained over-estimated were located in DPSs that permitted substantial travel in deep areas of the ocean (West Indies (DPS 1) and Southern Pacific (DPS 13) mainly).

### 4.4. The Particular Case of the Arabian Sea (DPS 14)

The blubber, where energy is stored, increased in all cases over the 2-year simulation period, but proportionally less than the rest of the soft body. The only exception was the growth of the individuals observed in the Arabian Sea, characterized by a decreasing blubber weight and sub-nominal total weight over the 2-year simulation period. The Distinctive Population Segment in the Arabian Sea did not exhibit significant migrations, as mentionedin the literature [74]. The endangered status of DPS 14 is, however, interpreted to be due the over-whaling performed by the Soviet Union in the 1960s that brought the population to a low viability [75], rather than other problems linked to the relative confinement of the individuals. In all cases, the results tend to support the consistency of the parameter estimates available in the literature, but they do not allow to address the problems of sub-population viability as they were summarized by Bettridge et al. [7]. However, the model constitutes a preliminary framework allowing to examine what is known and what is required to better understand the dynamics and changes in physiological conditions of whale individuals and to implement specific conditions that can degrade or improve these conditions.

### 4.5. Lessons Learned, Where Is More Information Needed?

The individual-based model formulated for this study is a minimal model designed to simulate the entire humpback whale population. It is based on the biological concept of a minimal individual possessing all the essential functions of a living whale. The two central components of the individual's dynamics are its behavior and physiology. The physiological processes correspond to unconstrained isometric growth. Individuals are assumed to have only a proximal and limited perception of their environment. From this small set of fundamental ecological principles, we have seen that it is possible to produce a plausible

picture of migration patterns, mimicking the whale "superhighways" reconstructed by the WWF from available recorded tracks [25].

There is nonetheless room for improvement and the perspectives for future developments are rich. First the complexity of the simulated environment should be augmented. Our framework is in fact designed to be coupled with a Global Ocean circulation model and it will be important to refine the individual and environmental scales. Secondly, a more complete physiological and behavioral description of whale states that includes reproduction is necessary. Finally, additional simulations with realistically distributed abundances among the 14 DPSs could be made to construct a set of statistics to treat simulated data. This could be used to estimate how many observations are needed to validate hypotheses.

## 5. Conclusions

The model presented here is the first one to quantify humpback whale dynamics within a virtual Global Ocean framework. Because whale population dynamics have been described as conditioned by individual behaviors (rather than demographic or population-level processes), the model used individual-based dynamics, driven by an ethogram, to simulate the seasonal activities of 420 whales over a 2-year period. It simulates synoptic states of the group of humpback whales for each DPS, including physiological state and displacements. The model is intended to help interpret the individual-level observations which predominate in whale studies. Hypotheses invoked about the role of environmental drivers on individual dynamics, which interact with behaviors and the physiology of the organisms, can be modeled and compared with observation data. It is possible to place whales in observed or simulated environmental dynamic fields (current, temperature, seismic, acoustic, forage fish and krill biomass . . . ), including human-induced disturbances (shipping, drilling, . . . ). As such, conceived as a collaborative tool, it is intended to evolve to support management and conservation initiatives.

**Author Contributions:** J.-M.G. and J.C.-G. contributed equally to all steps of this work. All authors have read and agreed to the published version of the manuscript.

**Funding:** This research received no external funding.

**Institutional Review Board Statement:** Not applicable.

**Informed Consent Statement:** Not applicable.

**Data Availability Statement:** Not applicable.

**Acknowledgments:** The idea for this work originated with a starting grant program on whale bioacoustics funded by the CNRS-INEE (France) to the first author, under the acronym C-SCAPE (2016).

**Conflicts of Interest:** The authors declare no conflict of interest.

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
