# Peer review of "A First Individual-Based Model to Simulate Humpback Whale (Megaptera novaeangliae) Migrations at the Scale of the Global Ocean"

_jmse, doi:10.3390/jmse10101412_

Round 1

Reviewer 1 Report

This MS provides an individual-based model of humpback whale migration.  Although this is a good foundational idea, the current version of the paper is difficult to follow, and some of the contentions in the MS do not make sense (or are unclear).  Further, the abstract suggests that there is disagreement about whether migration in whales concerns fitness, which is overly simplified and not exactly correct.  Due to the lack of data, this MS does not contribute very much to arguments about individual decisions on migration, given the lack of data about individuals and their migration routes.  However, the prediction of the model that whales generally are following loxodromic routes, however, could be of interest to a wide audience.

This paper also ignores some hypotheses about the whale migration and does not cite some of the more influential papers in this area of research.  In general, the coverage and discussion of previous research in the field should be integrated into this paper.  For example, the authors ignore work by Pitman et al. (2019, DOI 10.1111/mms.12661) suggesting that whales may migrate for molting. 

Another useful reference on the role of resource-tracking and memory in whale migration:  https://doi.org/10.1073/pnas.1819031116

And one on adaptation to environmental change with regard to blue whale migration:

https://doi.org/10.1038/s41598-020-64855-y

The second paragraph of the Introduction is not clear.  For example, it is unclear why the authors think the status of whale migration (pattern versus process) is unclear; these are two different ideas, but clearly migration is a process that has particular patterns, at least in the usual sense of these words.  If the authors are using a different sense, it would help if they defined their understanding of these terms clearly. 

The MS should be edited by a native English speaker for clarity, a process which would facilitate future reviewers’ efforts.

Lines 116-122.  This is unclear and very difficult to read.  Please separate the components, explain each clearly, and avoid run-on sentences.  What is bones’ growth regulation rate, for example? 

L 124.

What exactly is the ‘total optimal weight of the whale,’ and how was that determined?

L 130. The paragraph starting here does not make sense and I cannot figure out what the authors meant.

The weight loss rate uM is decomposed as the sum of the terms corresponding to the basal metabolic cost, ub, to the metabolic cost of swimming, um, to the cost of interacting, ui and to the cost of feeding uf.

L. 158-164.  This section is unclear.  What is a ‘proper’ behaviour, and why exactly is migration not considered to be one?

L. 172-3.  The ‘arbitrary threshold parameter’ should be clarified.

L. 187, 190-191.  Reword for clarity.

L. 204.  “Socializing” behavior does not ‘interact’ with other behaviors in this sense.  Rephrase for clarity.

Ll. 452-454 “and individual behaviors predominate over demographic processes for the control of the population dynamics.  When applying these criteria….”

What is the basis for this statement? In general, demographic processes, in particular the dramatic declines in whale populations, have a very large influence on population dynamics.  Also, the criteria to which the authors are referring are unclear.

The first paragraph of the Discussion contains a great deal of material that should be included in the Introduction.  It also needs revision, as it is very difficult to read in its current form, in part because it presents a longish list of facts without integrating, discussing, or linking them.

Typos/grammar

Abstract

Lines

1 ‘Whales’ migrations’ or ‘whale migrations’

36. What are ‘the two tropics?’  The equator does not seem like a good dividing line for whales (or other species) whose populations cross that latitude.

52.  [7] cannot be the subject of a sentence.

69 & 72   

86  individual humpback whales rather than ‘humpback whales individuals’

90 ‘bedrock’ rather than ‘bedrocks’

92 ‘formats’ rather than ‘format’ and punctuation marks go inside quotation marks.

113-114 – What does this mean?  The length was ensured in all conditions?

118. ‘bones accretion rate’ should be ‘the accretion rate of bone’ or ‘bone’s accretion rate’

123.  Lmin not Lmin

125.  ‘dynamics of bone weight’ rather than ‘dynamics of bones weights’

155.  individual whales rather than ‘whales individuals’

156.  ‘with movement’ instead of ‘to movement’

158.  that the term ‘migrations’ refers to….

172.  Check for subject-verb agreement (transition…are)

177.  Subject-verb agreement.  Change to ‘breeding, which was considered….’

179.  Change in ‘likelihood of occurrence.’

222.  Define ‘cape’ as used here.

225.  Correct ‘an average directions’

451.  Correct ‘whales individuals.’

Table 1

Resting – ‘low level of activity’  would be more clear. ‘Movements’ should be singular here.

Exploring – The phrase ‘series of diving’ does not make sense.  Also we cannot know the purpose of each dive, so the latter part of the description may not always be correct.  Add ‘generally’ before ‘to gather….’

Breeding – Delete ‘Socializing.’

Socializing – Interacting either distantly (…) or ….

            This category should be renamed if the authors prefer to include agonistic behaviors.  It could, for example, be called ‘Social Behavior’.

Author Response

Dr. J. Coston Guarini and I would like to thank you for the time you spent and the valuable comments you made to help us improving the manuscript. You will find a reply for each of your comments and questions in the attached file.

Reviewer 2 Report

The manuscript (MS) was designed to estimate movement of different life purposes of the whale modeling energy budget based on the size, and is first attempt to model the behavior. 

However, the writing language is needed to be reviewed by a professional editage. There are so many unnecessary mark "comma" in the sentences, which make the sentences longer and misunderstanding. The writing syntax are a lot there, . font of the variable units is italic and in some cases regular font. The authors should take care the MS by reviewing such errors in the text looking at the author instruction of the journal JMSF again. There are missing number of Table missing e.g. (Table ??) or (Figure 1 1) repeated number. There are many syntax errors in the text when taking it in to account as whole, meaning there are no standard format for the same info, changing in the text time to time, e.g. some cited sentences. In some Figures, axis labels (not title) not completed cutting off. In Figure 1, font size in flow chart of the model could be larger for better seeing them. In Figs, Julian day is guessed to be cumulative days continuing from one to the next year, since the Julian day exceeds 365. So what is initial year of the Julian days, or there is no the year. 

In the discussion, limitations of the model developed could be presented if available. In the results, or Material and Methods, there are some assumptions before applying the model. 

The MS needs minor changes and revions for better understanding.

Author Response

(The authors gave the same response as above.)

Round 2

Reviewer 1 Report

The well-crafted revision fully addressed the substantive comments.